# FARV: Leveraging Facial and Acoustic Representation in Vocoder For Video-to-Speech Synthesis

## Abstract

In this paper, we introduce FARV, a vocoder specifically designed for Video-to-Speech (V2S) synthesis, which integrates both facial embeddings and acoustic units to generate speech waveforms. By sharing the acoustic unit vocabulary in our two-stage V2S pipeline, FARV effectively bridges the domain gap between the visual frontend and the vocoder without requiring finetuning. Furthermore, by embedding visual speaker images into the acoustic unit representations, FARV enhances its ability to preserve speaker identity. Experimental results demonstrate that FARV achieves leading scores in intelligibility and strikes a favorable balance between speaker characterisitcs preservation and acoustic quality, making it well-suited for practical V2S applications[1].

## 1 Introduction

Video-to-Speech (V2S) synthesis (Prajwal et al., 2020a) aims to generate intelligible, natural-sounding speech directly from silent video inputs, leveraging visual cues such as lip movements and facial expressions for audio recovery. V2S is particularly useful in scenarios where only visual information is available to infer the speaker's speech, such as in silent video meetings or for individuals who cannot produce voiced sounds.

Most V2S approaches (Mira et al., 2022; Choi et al., 2023a; Hsu et al., 2023) rely on a two-stage framework: an upstream model that extracts audio representations (Mel spectrograms or acoustic units), followed by a vocoder that converts these representations into waveforms. Since the upstream model and the vocoder are trained separately, a domain gap often arises between these two stages because the vocoder is trained on ground-truth acoustic representations from clean datasets, which may not adapt well to the outputs of the frontend encoder. Therefore, vocoders play a crucial role in V2S systems, as they are responsible for bridging the gap to upstream model outputs and recovering the audio at the same time. However, due to the drawbacks of vocoders, many existing V2S models still face significant challenges, particularly concerning the preservation of speaker identity (Hsu et al., 2023) and the generalization between synthesis stages.

Unit-based vocoders can mitigate the domain gap by sharing a common vocabulary of discrete units between the stages and can be adapted to the upstream model without concern. However, they often lose crucial speaker-specific information, resulting in synthesized audio that sounds less natural and fails to accurately reflect the original speaker's identity. In contrast, mel-based vocoders offer better speaker preservation by utilizing Mel spectrograms, which provide richer frequency details. Yet, their sensitivity to spectral differences limits their generalization across the two stages.

To overcome these limitations, we propose an approach that combines the generalizability of unit-based vocoders with the speaker-specific preservation capabilities of mel-based vocoders. Specifically, we leverage a pre-trained facial representation extractor (Zheng et al., 2022) to capture speaker characteristics from the speaker's image. This allows us to seamlessly integrate speaker-specific information into the vocoder. At the same time, the vocoder synthesizes audio based on acoustic units shared with the frontend encoder. This approach enables our V2S model to maintain the generaliz-

---

[1]Code and weights will be made publicly available upon acceptance of this paper.

ability of the frontend encoder while preserving essential speaker identity features, resulting in more natural and speaker-consistent speech synthesis.

The contributions of this work can be listed as follows:

1. We introduce FARV, a unit-based vocoder that integrates facial image embeddings and acoustic units for speech synthesis, which is specifically designed for V2S.

2. FARV addresses the limitation of unit-based vocoders that struggle to retain speaker characteristics, offering a balanced approach between preserving speaker identity and ensuring high acoustic quality in V2S synthesis.

3. We demonstrate that mel-based vocoders require finetuning on frontend encoder outputs to adapt effectively to V2S. In the meantime, FARV is more resilient and can be adapted to frontend encoder even in a zero-shot manner.

4. Our V2S method achieves leading performance in acoustic intelligibility, relying solely on visual input during both training and evaluation, underscoring its practicality for real-world V2S applications.

## 2 RELATED WORK

### 2.1 VOCODERS

In speech synthesis, it is important to reconstruct speech from a compressed latent acoustic representation. To meet this need, vocoders offer a way to convert the acoustic representations to waveform. In this way, any speech-related tasks can first train a model that generates the acoustic representation, which is then fed to vocoders to synthesize speech. Vocoders are widely used in text-to-speech (TTS) (Ren et al., 2022; Shen et al., 2018; Li et al., 2019; Du et al., 2022; Jia et al., 2019; Wang et al., 2023) and have also become a favorable choice for V2S. Common choices for latent acoustic representation for vocoders are the Mel spectrogram (Kong et al., 2020; Yamamoto et al., 2020; gil Lee et al., 2023) and the hidden unit (Polyak et al., 2021; Lee et al., 2022; Hsu et al., 2023).

While vocoders based on Mel spectrograms (mel-based vocoders) have the ability to retain speaker characteristics in speech synthesis, they are often vulnerable to domain shifts in speech conditions. For example, chun Hsu et al. (2020) tested mel-based vocoders on unseen speakers, and have found that all tested mel-vocoders suffer from a significant domain gap. This domain gap is probably caused by the over-sensitivity of vocoders to frequency distributions of input audio, which interfers with vocoders in domain adaptations.

Later, Lee et al. (2022) found that hidden units from HuBERT (Hsu et al., 2021) can serve as an acoustic codebook to resynthesize audio waveform. Inspired by this, ReVISE (Hsu et al., 2023) proposed to use unit-HiFiGAN (unit-based vocoder) in V2S and speech enhancement. However, Hsu et al. (2023) also mentioned that HuBERT units focus on speech content and only contain knowledge capable of reconstructing utterances of spoken sentences, neglecting speaker characteristics (speaker identity like gender or age).

### 2.2 VIDEO TO SPEECH(V2S)

Lip2Wav (Prajwal et al., 2020a) was a pioneering work of V2S that performs speech recovery on a self-made dataset. However, Prajwal et al. (2020a) found that mel-based neural vocoders perform poorly on their generated Mel spectrograms. VAE-GAN (Hegde et al., 2022) and VCA-GAN (Kim et al., 2022) proposed adversarial learning in V2S and gained comparable performance with other supervised methods. Later, SVTS (Mira et al., 2022) tested their V2S method on LRS3 Afouras et al. (2018) and VoxCeleb2 Chung et al. (2018), which still suffers performance degradation on unseen speakers. Following SVTS, IntelligibleL2S (Choi et al., 2023b) incorporated unit and mel as vocoder input for V2S. Then, MultiTask (Kim et al., 2023) and AccurateL2S (Hegde et al., 2023) incorporated additional textual information and speaker embedding to further enhance V2S performance. However, since textual and acoustic speaker embedding is not always available, their application in practical scenarios of V2S is limited. To eliminate the need for additional inputs, DiffV2S (Choi et al., 2023a) proposed a diffusion method conditioned on visual embedding.

## 2.3 UNSUPERVISED FACIAL REPRESENTATION LEARNING

Unsupervised facial representation learning (Bulat et al., 2022; Zheng et al., 2022) is able to provide prior knowledge about facial identity of a person that can be transferred to downstream tasks like face attribute recognition (e.g. gender or age). Paplham & Franc (2024) compared existing methods on facial gender estimation with a unified benchmark and have found that FaRL (Zheng et al., 2022) with an MLP significantly outperforms other methods thanks to the amount of data used in its pretraining. Similar to CLIP (Radford et al., 2021), FaRL is pretrained with image-text pairs on a human face subset of LAION-400M (Schuhmann et al., 2021) using contrastive loss. We therefore leverage FaRL in the training of unit-HiFiGAN to provide insights of visual speaker characteristics.

## 3 METHODOLOGY

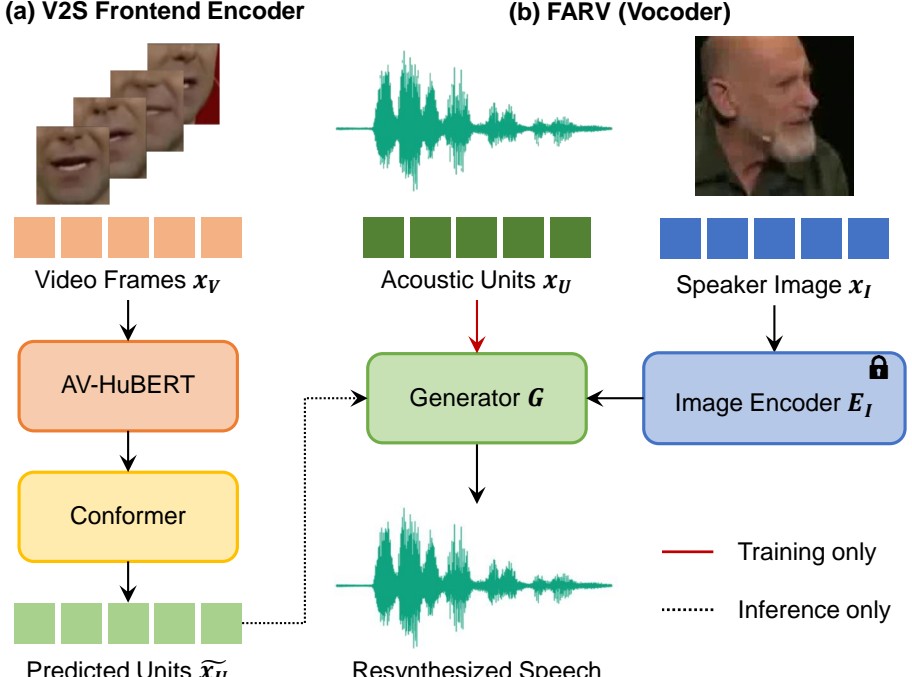

Figure 1: Overview of our V2S framework. (a) Our V2S frontend encoder is a pre-trained AV-HuBERT model followed by conformer. (b) We apply FaRL image encoder to provide visual speaker embedding for unit vocoder to preserve speaker characteristics. The proposed unit vocoder takes both acoustic units and visual embedding as input to synthesize audio.

## 3.1 FRAMEWORK DESIGN

Our V2S framework is a two-stage framework composing a visual frontend encoder and a vocoder. The frontend encoder utilizes a pretrained AV-HuBERT model (Shi et al., 2022a), followed by a Conformer module (Gulati et al., 2020). The AV-HuBERT model was shown to be crucial for efficient convergence in V2S tasks (Hsu et al., 2023), making it our preferred choice for the frontend backbone. To match the sampling rate of visual frames (25Hz) and that of acoustic units (50Hz), transposed convolution is applied to upsample the final output of frontend encoder. This frontend provides prior knowledge about audio-visual correlation to our V2S pipeline, offering accurate content recovery capacity.

The vocoder (FARV) is an adapted unit-HiFiGAN that combines visual embedding from the facial extractor(Zheng et al., 2022) and acoustic units. This audio-visual modality fusion will enable FARV to preserve speaker characteristics better. The acoustic units are clustered from the output of the pre-

trained HuBERT model (Hsu et al., 2021), which contains contextual information essential for the recovery of speech synthesis content. The unit vocabulary is shared for both frontend encoder and FARV.

## 3.2 FRONTEND ENCODER

Our frontend encoder is responsible for predicting acoustic units from silent visual inputs alone. Following ReVISE (Hsu et al., 2023), given $x_V$ as the silent video frames and $x_U$ as the acoustic tokens for ground-truth audio $x_A$, the V2S frontend is trained to produce predicted acoustic units, denoted as $f(x_V) = \widetilde{x_U}$. Cross-entropy loss is adopted to optimize V2S frontend, which is formulated as

$$L_{CE} = -\sum_j^C z_j \log softmax(f(x_V))$$

, where $C$ is the total number of classes of acoustic unit vocabulary. $z_j$ is the one-hot indicator sequence with the ground-truth label (the $j$th class) of the total $C$ acoustic classes representing $x_U$ for each unit. During inference, we simply take the argmax of the prediction for acoustic units $\widetilde{x_U}$ as the input to the vocoder.

To create an aligned setting for testing mel-based vocoders, we also train a modified mel-based version of the frontend encoder. In this version, we adjust the training loss of the proposed method to optimize the frontend encoder using an L1 regression loss

$$L_1 = -||M(x_A) - f(x_V)||_1$$

, where $M$ means Mel spectrogram conversion in logarithm. In this way, visual frontend $f$ learns to generate Mel spectrogram. We then apply a vanilla mel-based vocoder to transfer the encoder output to audio waveform. This V2S framework is made for comparsion and denoted as ReVISE (Mel).

## 3.3 FARV

Since unit vocoders struggle to gain speaker characteristics from acoustic units only, we apply FaRL image encoder (Zheng et al., 2022) as the visual speaker information extractor for unit-HiFiGAN to provide extra hints about speaker information. Specifically, the proposed vocoder takes both acoustic units and a visual frame cropped from the input video as input. We add the encoded image embedding from FaRL to unit embedding to provide visual guidance for speech generation. The image embedding is broadcasted to match the length of acoustic units.

For the proposed unit vocoder, given the speaker image input $x_I$ and acoustic units $x_U$ as input, the image encoder $E_I$ will encode $x_I$ to visual embedding $e_I$, while the lookup table for the acoustic unit vocabulary will map $x_U$ to unit embedding $e_U$ after convolution. During the entire training process, $E_I$ is frozen to only produce stable facial representation embedding for unit vocoder. Then, we add these embeddings sequentially to make the fusion of audio-visual modalities $p_{AV}$:

$$p_{AV} = e_I \oplus e_U \tag{1}$$

$p_{AV}$ is fed to generator $G$ to synthesize audio. Given discriminator $D$ (which is actually a set of discriminators (Kong et al., 2020)) and ground-truth waveform $x_A$, the adversarial training losses are defined as:

$$L_{adv}(D;G) = ||D(x_A) - 1||_2 + ||D(G(p_{AV}))||_2 \tag{2}$$

$$L_{adv}(G;D) = ||D(G(p_{AV})) - 1||_2 \tag{3}$$

Similar to HiFiGAN, besides adversarial loss, we also Mel spectrogram loss and feature mapping loss to ensure the fidelity of synthesized audio and stablize training:

$$L_{mel}(G) = ||M(G(p_{AV})) - M(x_A)||_1 \tag{4}$$

$$L_{FM}(G;D) = \sum_i^T \frac{1}{N_i} ||D^i(x_A) - D^i(G(p_{AV}))||_1 \tag{5}$$

where $T$ denotes the number of layers in discriminator and $N_i$ denotes the number of features on the $i$th layer.

The final optimization objectives for generator ($L_G$) and discriminator ($L_D$) are as follows, where $\lambda_{mel}$ is a hyperparameter for loss balancing set to 45 as in Kong et al. (2020):

$$L_G = L_{adv}(G; D) + L_{FM}(G; D) + \lambda_{mel} L_{mel}(G) \tag{6}$$

$$L_D = L_{adv}(D; G) \tag{7}$$

## 4 EXPERIMENTS

### 4.1 EXPERIMENTAL SETTINGS

#### 4.1.1 DATASETS

We applied LRS3-TED (Afouras et al., 2018) and LRS2-BBC (Afouras et al., 2022) datasets to test intelligibility of V2S systems and train the proposed vocoder. The splits of LRS3-TED are identical to that of Afouras et al. (2018). VoxCeleb2 (Chung et al., 2018) is also applied to testify equal error rate (EER) for speaker verification. All these datasets are also used to test the adaptation capability of vocoders.

We also use the audio-visual RAVDESS dataset (Livingstone & Russo, 2018) to test the capability of unit vocoders on gender and emotion classification. This is similar to the way Ji et al. (2024) tested their embedding but we only choose RAVDESS dataset of the benchmark (Livingstone & Russo, 2018) as it contains audio-visual resources. There are 8 emotions for classification[2] and they can be clearly reflected based on facial expression of the speaker.

#### 4.1.2 METRICS

For low-level detail reconstruction, we utilize Extended Short-Time Objective Intelligibility (ES-TOI) and Mel Cepstral Distortion (MCD) metrics, focusing on speech intelligibility and mel-cepstral differences, respectively. To assess audio-visual synchronization, we employ LSE-C and LSE-D metrics, following the implementation from Prajwal et al. (2020b); Chung & Zisserman (2016). For content accuracy, we evaluate Word Error Rate (WER) using a pretrained ASR system (Xu et al., 2020). The wav2vec 2.0 model and weights for ASR evaluation are sourced from `https://huggingface.co/facebook/wav2vec2-large-960h-lv60-self`, identical to the setup used in ReVISE (Hsu et al., 2023). For acoustic quality, similar to how Irvin et al. (2022) evaluates vocoder, we apply NISQA-MOS (Mittag et al., 2021) to give automated non-intrusive prediction of subjective mean opinion score (MOS) scores.

For speaker characteristics preservation, we apply Speaker Encoder Cosine Similarity (SECS) and Equal Error Rate (EER) as our metrics to evalutate speaker matching performance. Following Choi et al. (2023a), we use an off-the-shelf audio speaker encoder Jia et al. (2019) for the evaluation of speaker embedding and compute SECS. EER computation is similar to Shi et al. (2022b) for speaker verification, where the matching score is the cosine similarity of speaker embedding (Jia et al., 2019) for each pair of trials. We use only one clip for each pair's evaluation for simplicity. EER is always tested on VoxCeleb2, as speaker labels are available to construct test pairs for speaker verification.

#### 4.1.3 TRAINING DETAILS

For all frontend encoders, we adopt the training settings from ReVISE (Hsu et al., 2023) and train the encoders on 8 GPUs for a maximum of 45,000 updates per GPU. The models are chosen based on the lowest L1 loss in the Mel spectrograms or the highest classification accuracy for the mel or unit-based frontend encoders during validation. We train ReVISE and ReVISE (Mel) frontend encoder both on our AV-HuBERT+Conformer structure for fair comparison.

For the vocoders, we follow the training setting of Hsu et al. (2023) to train HiFiGAN and unit-HiFiGAN on the single-speaker LJSpeech dataset (Ito & Johnson, 2017) resampled at 16kHz. Since visual images are required, we train FARV on the audio-visual LRS3-TED and LRS2-BBC datasets

---

[2]Emotions include neutral, calm, happy, sad, angry, fearful, disgust, surprised.

respectively, resuming from the checkpoint of the unit-HiFiGAN trained on LJSpeech. Vocoder training is limited to a maximum of 400,000 updates across 8 GPUs, with checkpoints selected based on the lowest validation loss of the Mel spectrograms.

## 4.2 V2S SYNTHESIS RESULTS

We compare the proposed method with existing approaches in terms of acoustic intelligibility, quality, and preservation of speaker characteristics in V2S synthesis. Given that finetuning on new datasets can significantly compromise the acoustic quality of Unit-HiFiGAN (Section 4.3.1), we utilize only the Unit-HiFiGAN model trained on LJSpeech without finetuning it on LRS2-BBC and LRS3-TED for this analysis.

### 4.2.1 INTELLIGIBILITY

We present a comparison of baseline methods and the proposed approach for V2S in Table 1. Even when only visual input is provided, the proposed method outperforms most existing approaches on both the LRS3-TED and LRS2-BBC datasets, demonstrating its strong V2S synthesis capabilities. Notably, while approximately half of the compared methods rely on additional acoustic speaker embeddings or textual information as supervision, which introduces out-of-domain knowledge beyond visual cues in V2S training, our method consistently ranks among the top two across all evaluated metrics. Notably, our V2S method consistently outperforms ReVISE in terms of audio-visual synchronization (LSE-C and LSE-D) and low-level metrics (ESTOI and MCD), indicating superior synchronization and a closer resemblance to the original audio in speech synthesis.

### 4.2.2 QUALITY AND SPEAKER CHARACTERISTICS PRESERVATION

We conducted a comparison between the proposed method and ReVISE on acoustic quality and speaker matching in Table 2. The results demonstrate that while ReVISE achieves superior performance in acoustic quality, it falls short in preserving speaker characteristics during speech synthesis. In contrast, the proposed method excels in maintaining speaker characteristics. However, it is also important to strike a balance between speaker characteristics preservation and acoustic quality in V2S application. As discussed in Section 4.3.1, the acoustic quality of ReVISE deteriorates significantly while gaining improvement in speaker matching metrics after finetuning on new datasets, which makes it less balanced in these two metrics.

## 4.3 VOCODER ADAPTATION CAPABILITY

Since the V2S frontend encoder is optimized to align with the expected inputs of the vocoder, the overall synthesis performance in the V2S framework is largely constrained by the vocoder's capabilities. Mel-based vocoders often face challenges in adapting to the frontend encoder's output, whereas unit-based vocoders effectively bridge this gap due to the shared acoustic unit vocabulary that is consistent throughout the frontend training.

In the following sections, we will explore vocoder adaptation for both datasets and V2S frontend applications. Dataset adaptation refers to the application of vocoders to new datasets that differ from the original training data, either through fine-tuning or in a zero-shot manner. We will examine the percentage drop in performance from the fine-tuned state to the zero-shot state, with smaller drops indicating a vocoder's stronger adaptability to new datasets. V2S adaptation, on the other hand, focuses on evaluation of vocoders when they are applied to the output of V2S frontend encoder, where domain gap may arise. We will assess the percentage drop in performance when vocoders are applied in a zero-shot manner to the V2S encoder output, highlighting their practical suitability for V2S applications.

In the following experiments, we perform finetuning for up to 400,000 updates on a single GPU, selecting the model checkpoint based on the lowest validation loss of the Mel spectrogram during vocoder training. FARV is further trained on the LRS3-TED dataset described in Section 4.1.3, as visual images are necessary for training the proposed vocoder. In contrast, other zero-shot vocoders are trained on the LJSpeech dataset.

| | | Sync | | Low-Level | | Cont. |
|---|---|---|---|---|---|---|
| **LRS3-TED** | | | | | | |
| **Method** | **Vocoder** | **LSE-C↑** | **LSE-D↓** | **ESTOI↑** | **MCD↓** | **WER↓** |
| *Methods taking visual-only input* | | | | | | |
| VCA-GAN | HiFi-GAN | 4.54 | 9.63 | 0.207 | 8.85 | 95.9 (Choi et al., 2023a) |
| DiffV2S | HiFi-GAN | 7.28 | 7.27 | 0.284 | 9.35 | 39.2 |
| Multi-Task | HiFi-GAN | 4.85 | 9.15 | 0.240 | 10.16 | 74.8 (Choi et al., 2023a) |
| SVTS | HiFi-GAN | 7.08 | 7.04 | 0.244 | 8.60 | 81.9 (Choi et al., 2023a) |
| *Methods requiring non-visual information* | | | | | | |
| VAE-GAN † | - | 2.06 | 8.26 | 0.15 | - | - |
| AccurateL2S‡ | BigVGAN | 7.89 | 6.85 | 0.37 | - | - |
| Multi-Task‡ | HiFi-GAN | 5.19 | 8.89 | 0.268 | 9.89 | 65.8 (Choi et al., 2023a) |
| SVTS† | HiFi-GAN | 6.04 | 8.28 | 0.271 | 8.02 | 78.0 (Choi et al., 2023a) |
| *Our Implementation* | | | | | | |
| ReVISE | Unit-HiFiGAN | 7.14 | 7.19 | 0.291 | 10.68 | **35.67** |
| Proposed | FARV | **7.45** | **6.89** | **0.299** | **8.38** | 36.81 |

| | | Sync | | Low-Level | | Cont. |
|---|---|---|---|---|---|---|
| **LRS2-BBC** | | | | | | |
| **Method** | **Vocoder** | **LSE-C↑** | **LSE-D↓** | **ESTOI↑** | **MCD↓** | **WER↓** |
| *Methods taking visual-only input* | | | | | | |
| VCA-GAN | HiFi-GAN | 2.63 | 11.61 | 0.134 | 9.35 | 101.1 (Choi et al., 2023a) |
| DiffV2S | HiFi-GAN | 7.51 | 9.81 | 0.283 | 9.85 | 52.7 |
| Multi-Task | HiFi-GAN | 7.19 | 7.01 | 0.322 | 10.22 | 61.0 (Choi et al., 2023a) |
| SVTS | HiFi-GAN | 7.87 | **6.30** | 0.301 | 7.97 | 76.6 (Choi et al., 2023a) |
| *Methods requiring non-visual information* | | | | | | |
| VAE-GAN† | - | 2.51 | 8.16 | 0.17 | - | - |
| AccurateL2S‡ | BigVGAN | 8.08 | 6.59 | 0.47 | - | - |
| Multi-Task‡ | HiFi-GAN | 6.88 | 7.32 | 0.341 | 9.37 | 57.8 (Choi et al., 2023a) |
| SVTS† | HiFi-GAN | 7.80 | 6.47 | 0.331 | 6.86 | 71.4 (Choi et al., 2023a) |
| *Our Implementation* | | | | | | |
| ReVISE | Unit-HiFiGAN | 7.48 | 6.79 | 0.300 | 11.05 | 37.65 |
| Proposed | FARV | **7.92** | 6.34 | **0.331** | 7.91 | **34.75** |

Table 1: Intelligibility evaluation on the LRS3-TED and LRS2-BBC datasets. Top-1 and top-2 performances for methods using visual-only input are highlighted in bold and underlined, respectively. Methods marked with †require additional speaker embeddings during training, while those marked with ‡utilize both audio embeddings and supplementary textual information.

| | **LRS3-TED** | | **LRS2-BBC** | |
|---|---|---|---|---|
| **Method** | Match | Qual. | Match | Qual. |
| | **SECS↑** | **NISQA-MOS↑** | **SECS↑** | **NISQA-MOS↑** |
| ReVISE | 53.93 | **4.10** | 52.31 | **3.98** |
| Proposed | **61.23** | 2.76 | **62.34** | 2.31 |

Table 2: Speaker matching scores for evaluating speaker characteristics preservation on the LRS2-BBC and LRS3-TED datasets.

### 4.3.1 DATASET ADAPTATION AND FINETUNING OF VOCODERS

In this section, we present the percentage of performance degradation observed in unit-based and mel-based vocoders when adapting to different datasets. The evaluation is conducted under both zero-shot and finetuned conditions to assess the generalization capabilities of these vocoders across various datasets.

We can observe from Table 3 that unit-based vocoders exhibit generally more stable performance on low-level metrics compared to mel-based vocoders in a zero-shot scenario, highlighting the inherent consistency of acoustic intelligibility in unit-based vocoders across different datasets. However, because unit-HiFiGAN is trained on a single-speaker LJSpeech dataset and encodes only the acoustic information relevant to speech content, it demonstrates the worst speaker matching performance in the zero-shot scenario. Additionally, as illustrated in Figure 2, unit-HiFiGAN experiences a signif-

icant decline in acoustic quality when finetuned on new datasets. This degradation may be due to its unit vocabulary, which is effective for preserving acoustic content but insufficient for encoding diverse speaker information.

FARV outperforms vanilla unit-HiFiGAN across all metrics, particularly in preserving speaker characteristics, with the only exception being acoustic quality in zero-shot settings. However, when both FARV and unit-HiFiGAN are finetuned on a new dataset, FARV consistently achieves better performance. As shown in Figure 2, the low drop rates in performance further demonstrate FARV's superior generalizability.

| | | **Vocoder Input: GT Acoustic Representation** | | | | |
|---|---|---|---|---|---|---|
| | | Match | | Qual. | Low-Level | |
| **Finetuned** | **Vocoder** | **SECS↑** | **EER↓** | **NISQA-MOS↑** | **ESTOI↑** | **MCD↓** |
| | | **LRS2-BBC** | | | | |
| | HiFiGAN | **94.71** | **20.62** | **2.81** | **0.863** | **1.57** |
| √ | Unit-HiFiGAN | 61.96 | 37.26 | 2.21 | 0.412 | 7.66 |
| | FARV | 65.16 | 30.52 | 2.35 | 0.464 | 7.25 |
| | HiFiGAN | **87.84** | **24.02** | 2.47 | **0.805** | **2.29** |
| × | Unit-HiFiGAN | 52.69 | 41.08 | **4.07** | 0.417 | 10.24 |
| | FARV | 60.23 | 28.36 | 2.64 | 0.440 | 7.94 |
| | | **VoxCeleb2** | | | | |
| | HiFiGAN | **95.67** | **19.76** | **2.94** | **0.821** | **1.62** |
| √ | Unit-HiFiGAN | 61.98 | 37.29 | 2.29 | 0.374 | 7.93 |
| | FARV | 63.94 | 29.28 | 2.48 | 0.393 | 7.57 |
| | HiFiGAN | **84.26** | **24.02** | 2.34 | **0.740** | **2.53** |
| × | Unit-HiFiGAN | 48.18 | 41.08 | **4.16** | 0.372 | 10.75 |
| | FARV | 60.95 | 28.36 | 2.87 | 0.400 | 7.92 |

Table 3: Dataset finetuning results where zero-shot vocoder adaptation is compared with its finetuned counterpart on the LRS2 and VoxCeleb2 test sets.

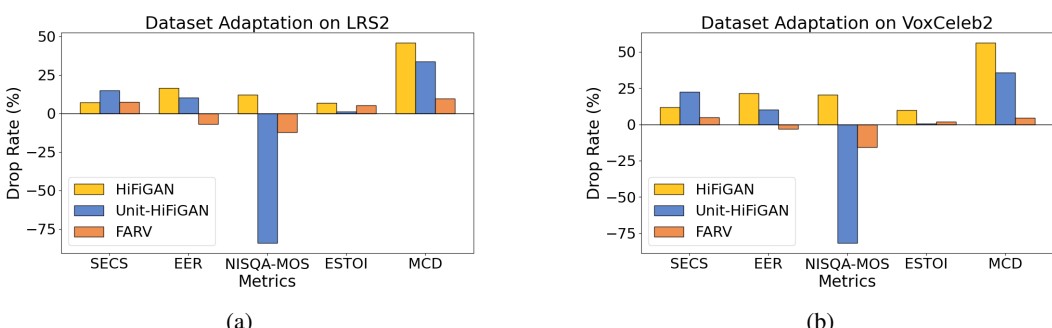

Figure 2: Performance drop rates of dataset adaptation on (a) LRS2-BBC dataset; (b) VoxCeleb2 dataset. The drop rates are presented as relative percentages, highlighting the domain gap between finetuned and zero-shot vocoders. Positive values indicate a performance drop, while negative values demonstrate that zero-shot performance is superior to that of the finetuned model.

### 4.3.2 FRONTEND ADAPTATION OF VOCODERS IN V2S

Since vocoders are ultimately used with frontend encoders for V2S applications, we explore the adaptability of different vocoders when used in conjunction with the frontend encoder. We present the evaluation of audio quality generated by vocoders that have not been finetuned on the frontend encoder output, using this as a baseline to assess their ability to recover audio from the frontend encoder within the V2S framework.

Table 4 and Figure 3 show that mel-based vocoders experience a much more significant drop in performance compared to unit-based vocoders when adapting to V2S frontend encoders. Specifically, metrics related to speaker matching, quality, and low-level features degrade significantly regardless

of whether the vocoders are finetuned on the LRS2-BBC dataset. This is likely due to frequency domain differences. As a result, the advantages mel-based vocoders demonstrate during training with ground-truth inputs are diminished in this scenario, highlighting the necessity of unit-based vocoders in V2S applications, especially when finetuning vocoders is not feasible.

| | | Match | | Qual. | Low-Level | |
|---|---|---|---|---|---|---|
| **Finetuned** | **Vocoder** | **SECS↑** | **EER↓** | **NISQA-MOS↑** | **ESTOI↑** | **MCD↓** |
| | HiFiGAN | 57.52 | 33.67 | 1.00 | **0.376** | **7.53** |
| × | Unit-HiFiGAN | 52.31 | 42.53 | **3.98** | 0.300 | 10.88 |
| | FARV | **58.57** | **30.01** | 2.64 | 0.311 | 8.75 |
| | HiFiGAN | 58.83 | 34.47 | 1.06 | **0.375** | **7.59** |
| √ | Unit-HiFiGAN | 59.88 | 37.59 | 2.28 | 0.312 | 8.68 |
| | FARV | **63.08** | **31.30** | **2.30** | 0.330 | 8.15 |

*Table 4: Frontend adaptation results when paired with the frontend encoder for V2S synthesis on the LRS2-BBC dataset. Vocoders labeled as "Finetuned" are trained on ground-truth audio from the LRS2-BBC dataset rather than the predicted outputs from the frontend encoder.*

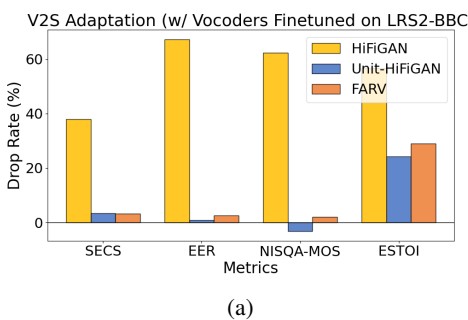
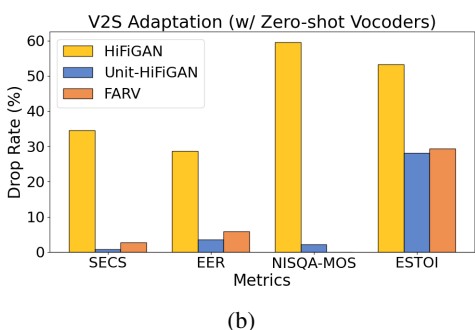

(a)                       (b)

*Figure 3: Performance drop rates of V2S adaptation evaluated on the LRS2-BBC dataset. The vocoders used are (a) finetuned on the LRS2-BBC dataset and (b) applied in a zero-shot manner. The drop rates are presented as relative percentages, comparing the ground-truth acoustic representation to the predictions from the V2S frontend encoder, which serve as input for the vocoders.*

### 4.3.3 FINETUNING ON FRONTEND OUTPUT FOR MEL-BASED VOCODERS

| | Sync | | Match | | Low-Level | | Qual. | Cont. |
|---|---|---|---|---|---|---|---|---|
| **Updates** | **LSE-C↑** | **LSE-D↓** | **SECS↑** | **EER↓** | **ESTOI↑** | **MCD↓** | **NISQA-MOS↑** | **WER↓** |
| 0k | 7.14 | 7.13 | 53.91 | 32.31 | **0.333** | **7.73** | 1.07 | 35.45 |
| 200k | **7.86** | 6.56 | 60.76 | 30.01 | **0.333** | 7.94 | 2.62 | **34.61** |
| 500k | 7.83 | **6.54** | **60.99** | **29.14** | 0.326 | 7.77 | **2.73** | 35.68 |

*Table 5: Effect of the mel vocoder (HiFiGAN) finetuned on Mel spectrograms generated by the trained ReVISE (mel) model in the LRS3-TED dataset. The differences indicate the number of finetuning updates applied to the vocoder, where "0k" signifies that HiFiGAN is trained on LJSpeech without any finetuning (zero-shot)*

In Section 4.3.2, we can observe that HiFiGAN incurs a significant drop in performance when adapted to the V2S frontend encoder in a zero-shot manner. Therefore, it is necessary to finetune HiFiGAN on Mel spectrograms generated by a fully trained frontend encoder to help it adapt to the domain gap. The results in Table 5 reveal a substantial improvement in acoustic quality after finetuning the vocoder on Mel spectrograms generated from the encoder output. Metrics for speaker matching and audio-visual synchronization also improve after finetuning, indicating a considerable performance gap induced by the domain gap for HiFiGAN when adapting to the V2S frontend encoder. Since practical V2S applications require converting visual inputs into speech, where audio is

often unavailable after deployment, this poses a significant limitation for using mel-based vocoders in V2S.

Therefore, while the mel-based vocoder maintains favorable performance across many metrics on different datasets (as shown in Table 3), it is still affected by the domain gap in V2S, making fine-tuning on frontend encoder outputs necessary for practical use. In contrast, unit-based vocoders can be applied to V2S in a zero-shot manner without requiring finetuning on model outputs, as they only predict acoustic units from a shared vocabulary used during the training of both the vocoder and the V2S frontend encoder.

## 4.4 EMBEDDING CAPABILITY

To testify the capability of the proposed method against traditional unit-HiFiGAN, we conducted experiments on unit embedding of these two vocoders. We train a simple baseline where a linear classifier is required to perform classification on gender and emotion. The linear model takes the output of unit embedding as input. During the entire training process of the classification baseline, the vocoder remains frozen and only provides embedding outputs to feed the linear classifier.

| Task | Model | Micro Acc (%) |
|---|---|---|
| Emotion | FARV | 69.44 |
| | Unit-HiFiGAN | 45.14 |
| Gender | FARV | 100.00 |
| | Unit-HiFiGAN | 81.94 |

Table 6: Micro accuracy for emotion and gender tasks.

Table 6 shows the results of the linear classification given different embeddings of the vocoder as input. When unit embedding of the proposed method are given, the accuracy of gender or emotion classification improves significantly compared to its unit-only counterpart. Notably, gender classification archives 100% accuracy for the proposed method, while unit-HiFiGAN fails to eliminate the ambiguity of speaker gender identity.

## 5 CONCLUSION

In this paper, we introduced FARV, a vocoder specifically designed for Video-to-Speech (V2S) synthesis that effectively integrates audio-visual modalities. Through a comparative analysis with existing vocoders, we identified their key limitations: mel-based vocoders struggle to adapt to the outputs of V2S frontend encoders, which limits their practical applicability, while unit-based vocoders face challenges in balancing speaker identity preservation with acoustic quality. FARV addresses these issues by incorporating facial image embeddings, which enhance the preservation of speaker characteristics, and by utilizing a shared unit vocabulary that seamlessly integrates with the V2S pipeline. Experimental results demonstrate that FARV achieves superior intelligibility and strikes an advantageous balance between preserving speaker characteristics and maintaining sound quality, even when adapted to new datasets. Overall, FARV shows significant potential for practical V2S applications, effectively minimizing the performance drops typically observed in mel-based approaches.

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

## A  DETAILED EXPERIMENTAL SETUP

### A.1  AUDIO-VISUAL DATASET SETUP

For audio-visual datasets, we use the official splits of LRS2-BBC. As LRS3-TED does not have official validation set, we apply the validation split provided by AV-HuBERT (Shi et al., 2022a). All resource in audio-visual datasets is used (224 hours for LRS2-BBC and 433 hours for LRS3-TED) to train V2S framework.

## A.2 Configuration

K-Means clustering with 2000 categories on the third iteration feature from the last layer of the HuBERT model Hsu et al. (2021) is applied to all our training dataset to build the acoustic unit vocabulary shared by both the frontend encoder and the vocoder. The HuBERT model for clustering is the BASE version pretrained on 960 hours of the LibriSpeech dataset Panayotov et al. (2015). The frontend encoder comprises AV-HuBERT LARGE of 325M parameters along with with a 4-block conformer module. The conformer has 4 attention heads with attention dimension of 256 and has 11M parameters for adapting visual input to acoustic representation prediction.

For the frontend encoder, we follow Shi et al. (2022a) to preprocess the visual input, as the upstream visual frontend uses an AV-HuBERT model. Preprocessing ensure that video frames are cropped to 96x96 based on facial keypoints detected by the dlib tool King (2009) and transformed into grayscale frames after an affine transformation. In acutal use of model, we further crop it to an area of 88x88 pixel region. During training, visual frames have a 50% chance of being horizontally flipped and are limited to 4.0 seconds from the start of each video. For evaluation, we consistently apply a center crop without flipping, and load the entire video into the model.

For FARV, we crop the speaker image at the middle frame of raw input video (unpreprocessed) in RGB and apply transformation identical to FaRL (Zheng et al., 2022) which applies center-crop to 224x224 to input RGB image after bicubic interpolation. We then normalize the image in RGB which is ready as the input to FaRL image encoder. No facial crop is performed as FaRL is trained on LAION-Face (Zheng et al., 2022), a dataset that contains human facial image-text pairs with the human face showing up at varied positions of different angles in image, which brings it the capability of zero-shot adaptation.

For Mel spectrogram generation during vocoder training and the ReVISE (Mel) frontend encoder, we follow van Niekerk et al. (2021), extracting 128-dimensional Mel spectrograms from raw audio at 10-ms intervals across all datasets. Since all our dataset has acoustic sampling rate of 16kHz, the hop size is set to 160, aligning with the upsampled visual features (100FPS after upsampling 4 times from 25Hz of original video clips). Size of fft is set to be 512.

## A.3 Training Setup

For frontend encoder, a tri-stage learning rate scheduler is applied with a max learning rate of 6e-5, which is identical to the training setting used in ReVISE (Hsu et al., 2023). AdamW optimizer is used with $\beta_1$=0.9 and $\beta_2$=0.98. We optimize the frontend encoder for at most 45k updates per GPU and freeze AV-HuBERT for first 5k updates of training to warm up the conformer module. We apply the batch size of 10 on each GPU.

For vocoder training, we apply the setup of van Niekerk et al. (2021) and train vocoders with AdamW optimizer with weight decay of 1e-5 ,$\beta_1$=0.8 and $\beta_2$=0.99. Exponential learning rate scheduler is applied with a decay rate of 0.999 for both the generator and the discriminators. Learning rate is set to be 1e-4. We apply the batch size of 8 on each GPU.

For vocoder training used for zero-shot scenario and frontend encoder training both take 8 RTX4090 GPUs to run for about 24 hours. Finetuning the vocoder takes only 1 GPU with identical settings to aforementioned setup.

## B Effect of Conformer Module

In addition to using the AV-HuBERT frontend encoder backbone from ReVISE (Hsu et al., 2023), we also incorporate a conformer module (similar to the approach in Mira et al. (2022)) following AV-HuBERT. This module helps to smooth the transition between the visual representations and the final prediction of acoustic units in the frontend encoder. To validate its effectiveness, we conducted a comparison (Table 7), which demonstrates performance improvements with the conformer. Based on these results, we include the conformer in our experimental setup.

| Models | Sync | | Match | Low-Level | | Qual. | Cont. |
| | LSE-C↑ | LSE-D↓ | SECS↑ | ESTOI↑ | MCD↓ | NISQA-MOS↑ | WER↓ |
|---|---|---|---|---|---|---|---|
| ReVISE w/o Conformer | 7.11 | 7.20 | 53.84 | 0.290 | 10.71 | 4.09 | 36.27 |
| ReVISE w/ Conformer | 7.14 | 7.19 | 53.93 | 0.300 | 10.68 | 4.10 | 35.67 |
| Proposed w/o Conformer | 7.42 | 6.91 | 61.31 | 0.298 | 8.40 | 2.74 | 37.51 |
| Proposed w/ Conformer | 7.45 | 6.89 | 61.23 | 0.331 | 8.38 | 2.76 | 36.81 |

Table 7: Effect of Conformer module for ReVISE and proposed method on LRS3-TED dataset.

