# OpenReview forum: "FARV: Leveraging Facial and Acoustic Representation in Vocoder For Video-to-Speech Synthesis"
_ICLR.cc/2025/Conference — ICLR 2025 Conference Withdrawn Submission_

### Official Review · Reviewer_NFDH · 2024-10-28

**Soundness:** 2
**Presentation:** 2
**Contribution:** 2
**Rating:** 3
**Confidence:** 5

**Summary:**

This paper introduce a vocoder, FARV, which integrates both facial embeddings and acoustic units for video-to-speech synthesis task. The FARV tackles the challenge faced in unit-based vocoder which struggles to retain speaker characteristics, offering a balanced approach between preserving speaker identity and ensuring high acoustic quality in video-to-speech synthesis. It also shows robustness in a zero-shot manner.

**Strengths:**

- The paper delivers extensive experimental results with ablation studies, showing the proposed model's robustness in a zero-shot manner.

**Weaknesses:**

- This paper does not seem novel; this work has a similar initial concept to [1] in that the proposed model utilizes visual information to preserve speaker identity to generate the output speech. Furthermore, the concept of utilizing visual information as speaker information is also similar to [2], which is not referred in the paper. More importantly, the proposed model develops the unit-HifiGan [3] by simply adding the existing image encoder FaRL [4], which is incremental in terms of novelty. I would encourage the authors to clarify what specific innovations their approach offers beyond combining existing components.

- In section 4.1.3, the authors mentioned that they train the proposed FARV on the audio-visual LRS3-TED and LRS2-BBC datasets. If so, how come the zero-shot performances of the FARV on LRS2-BBC in Table 3 are actually “zero-shot”? Did the authors train differently in this case? The authors should clarify this clearly.

- The analysis for the significant decline in acoustic quality (NISQA-MOS) of Unit-HifiGAN when finetuned in Figure 2 is not very convincing because the other metrics performances show improvement. Please provide more insight into why acoustic quality declines while speaker matching improves and discuss potential trade-offs between these metrics.

- Since the proposed module is a vocoder itself, the vocoders are ultimately used with different frontend encoders for V2S applications (lines 425 in the manuscript). To do so, the authors should conduct experiments on different frontend encoders (for different acoustic units from different encoders) with the proposed vocoder, FARV. While I understand the purpose of Section 4.3.2, I am not sure what the authors try to address in Section 4.3.2 from the experiment. What is V2S frontend prediction?


[1] Choi, Jeongsoo, Joanna Hong, and Yong Man Ro. "DiffV2S: Diffusion-based video-to-speech synthesis with vision-guided speaker embedding." Proceedings of the IEEE/CVF International Conference on Computer Vision. 2023.

[2] Hong, Joanna, Minsu Kim, and Yong Man Ro. "Visagesyntalk: Unseen speaker video-to-speech synthesis via speech-visage feature selection." European Conference on Computer Vision. Cham: Springer Nature Switzerland, 2022.

[3] Hsu, Wei-Ning, et al. "Revise: Self-supervised speech resynthesis with visual input for universal and generalized speech regeneration." Proceedings of the IEEE/CVF Conference on Computer Vision and Pattern Recognition. 2023.

[4] Zheng, Yinglin, et al. "General facial representation learning in a visual-linguistic manner." Proceedings of the IEEE/CVF conference on computer vision and pattern recognition. 2022.

**Questions:**

- During the inference time period, do all the speakers are different from the those in the training set, especially when reporting the performances in Table 1?
- In Table 1, what does (Choi et al., 2023a) mean after every WER performance?

---

### Official Review · Reviewer_9d9R · 2024-10-31

**Soundness:** 2
**Presentation:** 2
**Contribution:** 1
**Rating:** 3
**Confidence:** 4

**Summary:**

This paper proposes a model for video-to-speech (v2s) synthesis based on discrete units and facial embeddings, conducting partial experiments to validate the approach. However, many of its contributions appear to have been previously proposed and verified in other studies.

**Strengths:**

This paper provides a comprehensive supplementary exploration of the previous work and further demonstrates the conclusions of the previous work.

**Weaknesses:**

- Several ideas in this paper have already been widely explored: (1) Using acoustic discrete units (often more accurately referred to as semantic discrete units) to enhance speech reconstruction capabilities. [1] (2) Leveraging facial image embeddings to provide identity information for timbre reconstruction—a concept similar to representing speaker embeddings in vocoders but in a different form [2].
- The authors also claim that their model can perform zero-shot v2s, yet the results in Table 6 suggest that, on unseen samples, it only manages to achieve coarse-grained gender and general emotional reconstruction. This limitation likely stems from a lack of clear mapping between facial images and timbre, as the paper essentially uses image embeddings as in-domain speaker embeddings, which does not enable true zero-shot application. To demonstrate the value of image embeddings, the authors might consider an additional experiment comparing their approach to traditional speaker embeddings, highlighting whether facial images convey more information beyond speaker identity.
- Since v2s is fundamentally an audio synthesis task, subjective human ratings are often crucial in evaluations. However, the authors have not provided comprehensive MOS (Mean Opinion Score) results for the test set. Additionally, NISQA-MOS appears to be primarily a metric for assessing audio quality rather than speaker similarity, which may not fully suit this purpose, as seen in Table 2.

[1] Revise: Self-supervised speech resynthesis with visual input for universal and generalized speech regeneration. CVPR2023
[2] DiffV2S: Diffusion-based Video-to-Speech Synthesis with Vision-guided Speaker Embedding. ICCV 2023

**Questions:**

To demonstrate the value of image embeddings, the authors might consider an additional experiment comparing their approach to traditional speaker embeddings, highlighting whether facial images convey more information beyond speaker identity.

---

### Official Review · Reviewer_CmHq · 2024-10-31

**Soundness:** 2
**Presentation:** 3
**Contribution:** 2
**Rating:** 3
**Confidence:** 4

**Summary:**

This paper discusses a solution to the problem of mapping silent videos containing lip motion to accompanying and plausible speech sounds.  AV-Hubert is used to encode audio/visual speech to a joint vocabulary, and a vocoder is used to reconstruct a speech waveform from a corresponding sequence of these units.  Furthermore, to ensure speaker characters are preserved during synthesis, the generation is conditioned on a representation of speaker identity from a pre-trained visual model.  The results suggest that for the given dataset, the proposed approach beats the baseline.

**Strengths:**

The problem being tackled is interesting and challenging.

The main contribution, conditioning on speaker identity from the visual modality to preserve speaker characteristics, is a simple idea.  However, the approach seems to be effective given the results in the paper and the example video demonstrations provided.

Public data are used and the authors state that code and models will be made available.  This is useful for repeatability.

**Weaknesses:**

There is a number of objective metrics used to evaluate the approach against the baseline.  However, I am surprised that no subjective assessment has been included.  Human viewers are highly sensitive to discrepancies between the audio and the visual modalities of speech, and often artifacts to which we are sensitive are missed by objective metrics.

Only very short snippets of ~1—2s of speech are generated.  The sentences are not complete, and often both the beginning and end are truncated.  How well are speaker characteristics preserved over longer sequences?  This point specifically affects my soundness score.  I have further concerns around this (see Questions section).

**Questions:**

In the paper (on page 2) you mentioned that V2S methods that use textual information have limited practical use.  The system here seems constrained to produce only 1—2 seconds of speech, so is this also not a severe practical limitation?  If you could generate longer sequences by concatenating shorter sequences, would there be artifacts at the concatenation boundaries?  Would speaker characteristics be preserved across longer sequences if they are formed from shorter independent sequences?

Figure 1 shows that predicted units are used only during inference.  Are they not also used in training?  The Equations for L_{CE} and L_{1} would suggest predicted units are used given the righthand side of the sequences contains f(x_{v}).

What is the operator in Equation (1) denoting?  The text mentions that the image embedding is added to the unit embeddings so is this representation adding the static image embeddings to each element of the sequence of unit embeddings?   Does a simple addition make sense since the units of these embeddings are different?

Why were the stopping points for training/fine-tuning selected?  Was there some convergence guarantee to that point?

When you mention that ReVISE falls short in preserving speaker characteristics, can I clarify?  Do you mean that it does not preserve the characteristics of the generated voice, or that it does not preserve the characteristics of the voice of the actual speaker?  I assume you mean the former and not the latter, as the latter would imply predicting the voice of the speaker just from their appearance.

At the end of Section 4.3.3 you mentioned that fine-tuning is necessary for practical use.  Is this a significant problem?  Is the fine-tuning not just a one-off cost?

My biggest concern about this work is that the model produces generated sequences that almost perfectly match the ground-truth sequences word-for-word.  Typically a forensic lipreader would use conversational context, body gestures, facial expression, and so on and still only transcribe speech with an accuracy of ~30%.  Here you are able to cut short sequences from the middle of sentences, and with barely no context produce the sequence of units that perfectly map to the correct words.  Further, there is a lot of variation in speech acoustics that cannot be seen visually:  velar stops, voiced vs. voiceless sounds, nasality, etc.  Given how this non-visual articulation significantly impacts visual coarticulation of speech, I do not understand how the model is able to predict without longer context which unit to use when many might fit because the differences in the articulation of the sounds is where they cannot be seen.  It is for this reason that the YouTube series BadLipReading works:  a multitude of sounds fit the same sequence of lip movements, but your model remarkably seems to always predict the sequence almost perfectly.

*Nitpicks* (would not affect a decision)

The punctuation of the equations is off:  the comma should be immediately follow each of the first two unnumbered equations on page 4.

It seems odd that when highlighting the best and second best performing models in Table 1, those that use textual information are not included.  Maybe they are for reference-only, but they appear in the list as any other method but are ignored in the ranking.

There are a few typos throughout the paper, and so a careful read though should be done to catch these.

**Details Of Ethics Concerns:**

No ethics concerns.

---

### Official Review · Reviewer_mjRU · 2024-11-03

**Soundness:** 3
**Presentation:** 2
**Contribution:** 2
**Rating:** 5
**Confidence:** 4

**Summary:**

This paper introduces FARV, a unit-based vocoder that incorporates both facial embeddings and acoustic units for video-to-speech synthesis. The authors demonstrate through experiments that FARV effectively preserves speaker identity and mitigates the domain gap issue.

**Strengths:**

1. The integration of facial embeddings and a unit-based vocoder effectively preserves speaker identity in V2S and mitigates the domain gap challenge.
2. The experiments and analysis are thorough- Presentation Issues:
1.	The second paragraph of the Introduction (Lines 32-38) is not clear. It highlights the importance of vocoders but then contradicts this by discussing their drawbacks without specifying what these are. Additionally, the phrase "the generalization between synthesis stages" in line 40 is vague.
2.	In the Methodology section, Figure 1 is unclear, particularly the structure of the crucial Generator G, making it difficult to infer from the text alone how the three inputs are processed within the Generator. Additionally, although the text indicates that FARV employs a two-stage training process, this is not clearly depicted in the figure.
3.	The appendix mentions that AV-HuBERT is frozen during training, this module should be indicated with a freeze symbol in Figure 1.
4.	The inputs for the lips and face in Figure 1 seem to come from different speakers; the lips are without a beard, whereas the face has a beard.
5.	In Figure 1, are the Predicted Units only for inference? Do they not participate in the training of the Vocoder?
6.	Line 210 appears to be missing a verb (perhaps "employ" or "utilize").
7.	The quote mark in line 447("Finetuned") and line 476 ("0k") are facing the wrong direction.

- Concerns about Novelty:
1.	Extracting ID information from faces for voice synthesis tasks, such as in VisualVoice and Face2Speech, is not particularly novel for ICLR.
2.	The experimental results are not persuasive enough. Even utilizing multiple pretrained models like AVHuBERT, HuBERT, and FaRL for feature extraction, the performance improvements are not significant, with some metrics even underperforming (e.g., WER in LRS3, LSE-D in LRS2).
3.	The performance of Mel-Based vocoders after fine-tuning is significantly better, as shown in Tables 1 and 5.
.

**Weaknesses:**

- Presentation Issues:
1.	The second paragraph of the Introduction (Lines 32-38) is not clear. It highlights the importance of vocoders but then contradicts this by discussing their drawbacks without specifying what these are. Additionally, the phrase "the generalization between synthesis stages" in line 40 is vague.
2.	In the Methodology section, Figure 1 is unclear, particularly the structure of the crucial Generator G, making it difficult to infer from the text alone how the three inputs are processed within the Generator. Additionally, although the text indicates that FARV employs a two-stage training process, this is not clearly depicted in the figure.
3.	The appendix mentions that AV-HuBERT is frozen during training, this module should be indicated with a freeze symbol in Figure 1.
4.	The inputs for the lips and face in Figure 1 seem to come from different speakers; the lips are without a beard, whereas the face has a beard.
5.	In Figure 1, are the Predicted Units only for inference? Do they not participate in the training of the Vocoder?
6.	Line 210 appears to be missing a verb (perhaps "employ" or "utilize").
7.	The quote mark in line 447("Finetuned") and line 476 ("0k") are facing the wrong direction.

- Concerns about the work itself:
1.	Extracting ID information from faces for voice synthesis tasks, such as in VisualVoice and Face2Speech, is not particularly novel for ICLR.
2.	The experimental results are not persuasive enough. Even utilizing multiple pretrained models like AVHuBERT, HuBERT, and FaRL for feature extraction, the performance improvements are not significant, with some metrics even underperforming (e.g., WER in LRS3, LSE-D in LRS2).
3.	The performance of Mel-Based vocoders after fine-tuning is significantly better, as shown in Tables 1 and 5.

**Questions:**

1.	Why use HuBERT to extract Acoustic Units instead of AV-HuBERT, which has multimodal capabilities and might learn unified Units more effectively?
2.	In Table 3, why is the NISQA-MOS score for Unit-HiFiGAN so high in the zero-shot scenario?

---

### Note · Authors · 2024-11-18

I have read and agree with the venue's withdrawal policy on behalf of myself and my co-authors.